# Female Collegiate Dancers Body Composition, Macronutrient and Micronutrient Intake Over Two Academic Years: A Longitudinal Analysis

**DOI:** 10.3390/jfmk5010017

**Published:** 2020-02-26

**Authors:** Ann F. Brown, Samantha J. Brooks, Sawyer R. Smith, Joelle M. Stephens, Alexandria K. Lotstein, Chad M. Skiles, Christopher J. Alfiero, Melanie J. Meenan

**Affiliations:** Department of Movement Sciences, College of Education, Health & Human Sciences, University of Idaho, 875 Perimeter Drive MS 2401, Moscow, ID 83844-2401, USA; sbrooks@uidaho.edu (S.J.B.); sawyers@uidaho.edu (S.R.S.); step1021@vandals.uidaho.edu (J.M.S.); lots2725@vandals.uidaho.edu (A.K.L.); skil4827@vandals.uidaho.edu (C.M.S.); calfiero@uidaho.edu (C.J.A.); mmeenan@uidaho.edu (M.J.M.)

**Keywords:** dancers, dietary intake, aesthetic athlete, collegiate environment

## Abstract

Collegiate dancers face unique challenges to maintain a lean aesthetic, optimal diet, and a high-performance level due to the various stressors in college. The purpose of this study was to examine changes in body composition (BC) and diet over two years. Participants (*N* = 17, 19.6 ± 1.6 years) completed two laboratory sessions per semester. Sessions included height and weight, BC, dietary intake, and a health history questionnaire. Regardless of rigorous dance training and variations in the academic calendar, no significant changes in BC or diet were observed within semesters of over two years. BMI was normal (24.9 ± 4.1 kg/m^2^) with fat mass exceeding 30% at all timepoints. Fat mass was negatively correlated with carbohydrate, fat, and protein intake (g/kg/day; *r* = −0.291, *p* = 0.004; *r* = −0.372, *p* < 0.0001; *r* = −0.398, *p* < 0.0001; respectively). Energy intake was within the recommended daily allowance (2040 ± 710 kcal/day), however may be insufficient for an active dance population. Protein (1.1 ± 0.5 g/kg), carbohydrate (3.7 ± 1.6 g/kg), calcium (835 ± 405 mg/day), iron (17 ± 15 mg/day), and potassium (1628 ± 1736 mg/day) intake fell below recommendations for an active population. Alterations in dance training and the demands of the academic calendar may be contributing to suboptimal dietary intake and BC in female collegiate dancers.

## 1. Introduction

Dance is a demanding and vigorous performance sport that requires a high level of strength, endurance, and flexibility. Aesthetic sports, such as dance, place a strong emphasis on a lean body composition (BC), however this may vary depending upon the style of dance [1,2]. Lower fat mass (FM) has been previously reported in professional ballet dancers when compared to collegiate dancers, ranging from 8–28% FM [3,4]. The expectation of dancers to perform at an elite level while maintaining a lean aesthetic likely impacts dietary choices, nutrient timing, and eating behaviors and heightens the focus on caloric intake [5,6,7]. Emphasizing aesthetics in dance not only impacts dietary behavior but also exercise and training behaviors. However, enhancing physical fitness and performance capacity may not always be the primary reason for a dancer’s choice to increase exercise time or intensity. The focus likely stems from the desire to increase energy expenditure especially with the strong focus on the appearance of a lean physique. Additionally, collegiate dancers may be at risk for Relative Energy Deficiency in Sport (RED-S) syndrome predominately characterized by low energy availability (with or without disordered eating behaviors), menstrual irregularity, decreased bone mineral density as well as impaired physiological function [8]. The physical and technical demands of dance pose potential risks for injury and health complications if adequate timing of energy intake and macronutrients are not met during dance training and performances [9]. However, the impact of micronutrients on dancers’ overall health and performance remains unknown.

The ability of collegiate dancers to maintain both ideal aesthetics and optimal athletic performance is dependent upon adequate nutrition as well as nutrient timing [10,11]. Nutrient timing is an especially important consideration in a collegiate population due to the variations in the training schedule, which is based on the unique academic calendar. In particular, collegiate dancers must maintain a full academic course load in addition to dance training, rehearsals, and performances. Current macronutrient recommendations for dancers are 3–5 g carbohydrates/kg/day, 25% of total caloric intake as fat, and 1.2–1.7 g protein/kg/day [12]. However, for athletes who often reduce caloric intake, including dancers, the recommended amount of dietary protein to maintain lean soft tissue (LST) may be as high as 2.3 g/kg/day [4,13]. In other athletic populations attempting to optimize BC, protein intake as high as 3.4 g/kg/day has been shown to improve BC (increased LST and decrease FM) when compared to those consuming lower protein (2.3 g/kg/day) [14]. Although, it is challenging to propose the same dietary recommendations for dancers because they rarely engage in standard resistance training practices (i.e., weight training) like most other athletic populations. For dancers to achieve the desired aesthetic for their style of dance, protein intake should remain high (>1.7 g/kg/day) in order to minimize loss of LST, which could possibly lead to decrements in overall strength and performance capacity [5,10,15].

Due to the heightened focus on body weight in dance, BC can often be overlooked and may have detrimental health consequences. Weight classifications, such as overweight and obesity, are primarily characterized using BMI in clinical and epidemiological settings. However, BMI is calculated as weight (kg)/height (m^2^) and does not account for subcutaneous versus visceral adipose tissue (VAT), or BC FM, bone mineral density (BMD) or LST [16]. Therefore, individuals with a normal BMI (18.5–24.9kg/m^2^) may appear to be of normal weight while classified as obese (FM >30%), resulting in physiological implications on overall metabolic health, also known as normal weight obese (NWO) [16] Additionally, individuals who demonstrate greater VAT exhibit additional health complications compared to overall FM [17]. Therefore, NWO and VAT may be more specific measures of metabolic and cardiovascular health in female collegiate dancers, compared to BMI.

College students in general often engage in suboptimal dietary and exercise behaviors which may increase their likelihood of becoming overweight and obese [18,19]. The collegiate environment possesses many unique stressors for students including new autonomy of food choices, concern for the cost of food items, increased availability and consumption of fast foods, inadequate variety of food choices available, and frequently skipping meals [20,21]. These stressors of the college environment coupled with athletic expectations can lead to coping behaviors that are detrimental to dietary intake in the dance population [22]. Specifically, for collegiate dancers, the ability to maintain a diet that supports ideal BC and performance is increasingly difficult due to several factors including living arrangement, access to fresh nutritional food, and timing of meals around college courses and dance technique classes. Therefore, it may be difficult for collegiate dancers to maintain adequate nutrition for optimal BC within the collegiate environment [15]. Additionally, the academic calendar includes breaks during and between the two 16-week semesters (fall: August–December and spring: January–May, respectively) which may pose nutritional and training challenges due to alterations in dance training schedules. A typical school course load requires collegiate dancers to be in the classroom for 12–20 h per week, outside of dance training. During the fall semester, students have a one-week break over the Thanksgiving holiday (November) and during the Spring semester, students have a one-week break for spring break (March). Additionally, there is a three-week break between Fall and Spring semesters. Consequently, these challenges may alter dietary and exercise behaviors, BC, and disrupt dance training and overall performance.

The academic calendar and other variables unique to a collegiate setting create a demanding and stressful environment for collegiate dancers compared to professional dancers. Conversely, professional dancers spend extensive time training compared to collegiate dancers, due to increased demands of performance as well as lack of academic demands. However, schedule differences between collegiate and professional dancers has not previously been compared in the literature. Due to the variance in the academic setting, it is necessary to understand how the college environment may impact dancers’ BC, dietary behavior, dance training and overall health throughout their collegiate dance career. Given the relationship between diet and performance, longitudinal research is necessary to allow for a better understanding of collegiate dancers’ nutrient recommendations for collegiate dance specific training [13,23]. Therefore, the purpose of this study was to examine BC, and macronutrient and micronutrient intake in a female collegiate dance population over two academic years.

## 2. Materials and Methods

Seventeen female collegiate dancers (ages 19.6 ± 1.6 years), enrolled as a major (37 credit hours of dance courses and 16 credit hours of technique courses) or minor (25 semester credit hours of dance, technique, and choreography classes) in Dance at the University of Idaho, participated in this longitudinal study taking place during the 2017–2019 academic years. In order to enroll in this study, participants were required to be injury free and non-smoking for at least six months prior to this study’s commencement. Prior to participation, all dancers read and signed an informed consent explaining all potential risks and benefits of the study. The study was approved by the University of Idaho Institutional Review Board (RB: 17-170, 23 August 2017) in accordance with the Helsinki Declaration.

Participants visited the Human Performance Laboratory (HPL) twice per semester. Visits occurred during the first and last week of each academic semester. Visits during the 2017-2018 academic year occurred in August (V1), December (V2), January (V3), and May (V4). Additionally, visits during the 2018–2019 academic year occurred in August (V5), December (V6), January (V7), and May (V8). Upon arrival to the HPL, dancers completed a health history questionnaire including information on personal and family health history, previous injuries, current health concerns, menstrual history, and dance training history. Three-day food logs (2 weekdays and 1 weekend day) were turned in to record all food and beverages consumed. Participants were given detailed instruction on how to quantify food intake to improve accuracy of their recordings. Food logs were analyzed using Food Processor, (ESHA, 10.13.1, Salem, OR). The average energy intake (kilocalories), macronutrients, and micronutrients of the 3-day food logs was used for analysis.

Following food log submission, height and weight were measured using a stadiometer and digital scale (DETECTO, Apex-SH, Webb City, MO). BC was determined by dual energy X-ray absorptiometry (DXA; Hologic Horizon™, Marlborough, MA). One anteroposterior scan was performed with participants in the supine position according to the manufacturer’s instructions by a certified X-ray technician. Results were analyzed with APEX software, version 4.5.2.1 (Hologic Horizon™; Marlborough, MA). The quality analysis for the densitometer was conducted daily using a standard aluminum spine block (Hologic Phantom) provided by the manufacturer. LST and FM (% and kg), VAT (g) and appendicular skeletal muscle mass index (ASMI; kg/m^2^), were obtained for analysis.

Data were analyzed using SPSS v. 24. to report descriptive statistics as mean ± SD and range at each timepoint. Pearson’s correlations were used to determine the associations between carbohydrate, fat, and protein (g/kg/day) intake and FM (kg). To analyze within-semester changes and across-semester (longitudinal) changes, we used linear mixed-effects models fit in program R (R Core Development Team 2019) using the lmer package. A random effect of individual ID was incorporated, to account for some individuals joining or leaving the study at different points, which resulted in differing sample sizes between timepoints. Significance was set at *p* ≤ 0.05.

## 3. Results

Of the 23 participants recruited, 17 dancers enrolled in the study (Figure 1). Participant average age was 19.6 ± 1.6 years old with 9 ± 5 years ballet training, 4 ± 2 years modern training, and 1 ± 2 years of competitive dance training. Additionally, participants average training load was 11 ± 2 h of dance training per week and 7 ± 7 h of dance rehearsals per week. Dance training and dance rehearsals are described in Table 1. There were no significant differences in dance training or rehearsals across all timepoints. No significant changes (*p* > 0.05) were observed in BC, kilocalorie, macronutrient and micronutrient intake variables within semesters or over the two academic years.

### 3.1. Body Composition

BC characteristics over two academic years are shown in Table 2. LST, FM, VAT, ASMI, and BMD did not differ between timepoints. Additionally, BMI was within a normal range at every timepoint with FM exceeding 30% at all timepoints.

### 3.2. Dietary Intake

Energy intake and macronutrient data are shown in Table 3. Total kilocalories consumed was within the general recommended intake of 2000 kcals/day (2040 ± 710 kcal/day); however, nutrient intake was below recommendations for average carbohydrate intake (3.7±1.6 g/kg/day). Additionally, protein intake (1.1 ± 0.5 g/kg/day) was above the recommended daily allowance (RDA: 0.8 g/kg/day), however below recommendations for a dance population (1.2–1.7 g/kg/day) and other athletic populations engaging in weight loss (>2.3 g/kg/day) [12,13]. Average fat intake exceeded the recommended intake specific to dancers (<25% kcals/day) [11], as well as the RDA for overall health (<30% kcals/day) for all timepoints. Furthermore, when adjusting for body weight (kg), carbohydrate intake (g/kg), fat intake (g/kg), and protein intake (g/kg) were negatively correlated with FM. However, total energy intake (kcals) was not correlated with FM.

Micronutrients consumed are shown in Table 4. Participants consumed less than the RDA for fiber (RDA: 25 g/day) at all timepoints except visit 7, zinc (RDA: 8 mg/day) at all timepoints except visit 1 and visit 7, and iron (RDA: 18 mg/day) at all timepoints except visit 4. Additionally, participants consumed less than the RDA for vitamin D (RDA: 400 IU/day), calcium (RDA: 1000 mg/day), magnesium (RDA: 310 mg/day), and potassium (RDA: 4700 mg/day) at all timepoints. Furthermore, participants consumed greater than the RDA for sodium (RDA: 1500 mg/day) at all timepoints.

## 4. Discussion

The current study demonstrated that BC and dietary intake do not change over two academic years regardless of fluctuations in the collegiate dance training schedule. Energetic demands fluctuate over the course of the year due to periods of increased training and performance which requires increases or decreases in energy intake (nutrient timing) to maintain energy balance (i.e., caloric intake = caloric expenditure) to avoid physiological complications associated with RED-S syndrome. Thus, dietary intake that does not fluctuation with variations in training may result in unfavorable changes in body composition as well as decrements in training and performance if caloric demands are not properly timed with training, rehearsals and performances. However, the caloric demands of various dance styles are not well defined due to differences in energy expenditure [24]. In the present study, female collegiate dancers consumed adequate energy intake during some timepoints but did not meet the caloric demands during most timepoints, based on the general recommended 2000 kilocalorie diet. However, the RDAs are outlined to meet the needs of the average, predominately sedentary, individual and do not account for the increased physiological demands of training for athletic sports, such as dance. Therefore, macronutrient and micronutrient needs must be tailored to the type of sport, aesthetic demands, as well as athletic training loads. Energy intake may be indicative of fluctuations in dance training, performances, or academic demands. Furthermore, female collegiate dancers’ macronutrient and micronutrient intake did not meet the RDA for overall health, let alone an aesthetic athletic population. Consequently, collegiate dancer’s overall health and physiological function may be impaired causing unfavorable BC and dance performance may be compromised.

Similarly, to our current findings, other female collegiate dancer cohorts and adolescent dancers have reported inadequate consumption of folate, calcium, magnesium, and phosphorus [4,25]. In addition to reported low caloric intake in this population, it is also possible that the collegiate population relies on affordable and convenient processed food items that do not provide necessary micronutrients to support endocrine and bone health [25,26,27,28,29]. In the current study, only one dancer reported the use of a multivitamin to supplement micronutrients in the diet. To support optimal health, adjusting dietary intake to meet (possibly exceeded) RDAs would be ideal; however, a multivitamin would suffice if resources are lacking for dietary interventions for this population to prevent injury and adverse health effects. In a recent study, pre-professional contemporary female dancers demonstrated a negative energy balance during the week and positive energy balance over the weekend, indicating compensatory dietary behaviors [24]. The assessment of weekdays as well as weekend days independently, may give additional information into dancers’ energy intake patterns and future recommendations for nutrient timing over an academic semester to accommodate for weeks where training is the focus and weeks where rehearsal and performances are the focus. Furthermore, results from this longitudinal study indicate collegiate dancers train for a minimum of 25 h per week, and their BC and dietary intake remain unchanged. Overall, the lack of dietary changes with variations in training demands likely explains suboptimal BC findings. However, fluctuations in load or frequency of training were not large enough to elicit alterations in BC measures.

Macronutrient intakes were negatively correlated with FM, however total energy intake was not correlated with FM, which raises an interesting question regarding the overall impact of diet on dancers’ BC and health. Findings suggest that protein intake is negatively associated with FM, more specifically, those consuming lower protein in the diet had greater FM. Recent research suggests higher protein consumption (2.3 g/kg/day) can improve BC (increase LST and decrease FM) without observing changes in overall body weight in a female collegiate dance population [4]. Moreover, previous research supplementing with protein has also observed positive changes in BC in a wide variety of other athletic populations when protein intake is greater than >2.0 g/kg/day [12,27,29]. This level of protein intake is approximately double when compared to the habitual intake of those in the present cohort over all timepoints which may partly explain the concerning finding that the average BMI fell within a normal range (18.5–24.9 kg/m^2^), while average FM exceeded recommendations for the general female population (<30% FM).

For aesthetic sports, such as dance, the aesthetic focus is predominately placed on body weight not BC. Although the collegiate dancers in this study demonstrated a healthy weight (BMI < 24.9%), the average FM classification of obese (>30% FM) is concerning and classifies collegiate dancers as NWO, placing them an increased risk for metabolic syndrome and other comorbidities during adulthood. Although collegiate dancers may appear to achieve a lean aesthetic through weight maintenance, protein intake may not be sufficient to enhance BC. The combination of normal BMI classification and excess FM may have detrimental effects on overall health and risk for injury throughout the collegiate dance career [30]. Given the importance of diet to achieve a desired BC and level of performance, it is critical to further educate this population on how to achieve the desired aesthetic for their style of dance while also optimizing BC for performance and health benefits. This study has demonstrated that regardless of changes in training due to the academic calendar, dancers are not adjusting their dietary behavior to meet their fluctuating needs, ultimately negatively influencing BC.

Overall, this study yielded valuable descriptive BC and dietary intake of collegiate dancers, specifically regarding the challenges imposed by the academic calendar. Cumulative research and knowledge of collegiate dancers will provide a greater understanding of their BC characteristics, macronutrient and micronutrient needs and performance capacities to improve dietary health, decrease risk of injury and improve overall performance. Limitations of this study include a small convenient sample size, which made significant changes in descriptive variables difficult to detect. Additionally, 3-day food logs were self-reported, allowing for variability in the accuracy of food recorded. There is inherent bias with self-reported dietary intakes specifically underreporting “bad” foods and overreporting “good” foods. Furthermore, dietary intake over summer, winter, and spring breaks were not assessed. Dancers’ may demonstrate compensatory eating behaviors as well as a de-training period during these times, which may explain the lack of change in their BC over time. Our dance population was primarily trained in ballet and modern during the two-year study; therefore, our results should be compared to other collegiate dancers with a similar training background.

## 5. Conclusions

Despite rigorous dance training schedules and performances for collegiate dancers, no significant changes in BC or dietary intake over two academic years were identified. Therefore, dietary behaviors do not fluctuate with changes in dance training schedules. Furthermore, frequent disruptions to training schedules, alterations in the eating environment, and the demands of academic coursework may cause disturbances in collegiate dancers’ BC and dietary intake. BC may place dancers at risk for health complications with high FM and a healthy BMI and subsequently place them in the NWO classification. The health complications associated with excess body fat may be attributed to dietary choices and this should be a focus of future research on dancer health. Dietary interventions are necessary in collegiate dancers to identify proper nutrient timing strategies throughout the academic semester. This would help to achieve the RDA’s for macronutrients and micronutrients and obtain optimal diet quality to support health in a collegiate dance program.

## Figures and Tables

**Figure 1 jfmk-05-00017-f001:**
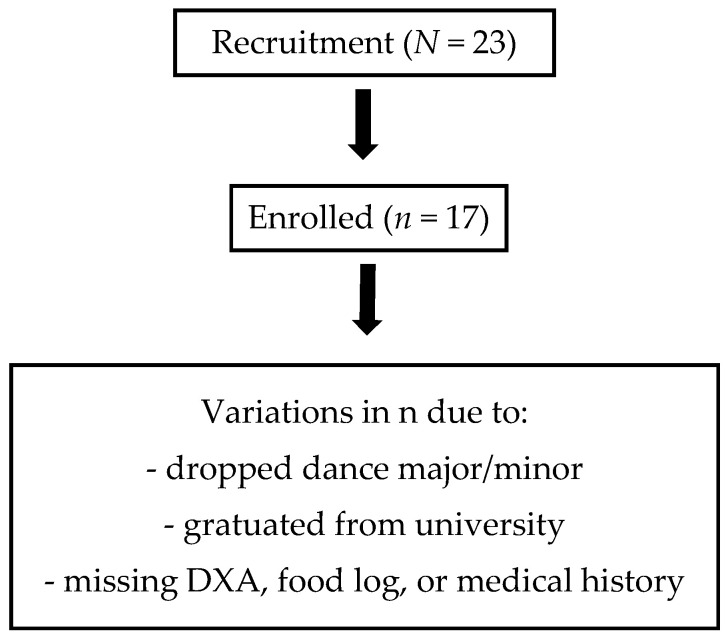
Variation in Participant Sample Size.

**Table 1 jfmk-05-00017-t001:** Dance training characteristics.

	Visit 1	Visit 2	Visit 3	Visit 4	Visit 5	Visit 6	Visit 7	Visit 8	*p*-Value
N	14	14	13	13	11	11	17	15	
Dance training (hrs/wk)	12 ± 4^1^ (6–18)^2^	12 ± 34 (6–18)	11 ± 5 (4–20)	13 ± 5 (4–20)	11 ± 8 (0–30)	11 ± 8 (0–30)	11 ± 6 (3–24)	10 ± 6 (3–24)	0.971
Rehearsals (hrs/wk)	8 ± 9 (0–36)	9 ± 9 (0–36)	10 ± 7 (0–19)	6 ± 7 (0–19)	9 ± 5 (2–15)	8 ± 5 (0–15)	7 ± 4 (0–12)	3 ± 4 (0–8)	0.081

^1^ Means ± SD. ^2^ Range. Hrs/wk, hours per week. Visits 1 and 5: August 2017 and 2018, Visits 2 and 6: December 2017 and 2018, Visits 3 and 7: January 2018 and 2019, and Visits 4 and 8: May 2018 and 2019.

**Table 2 jfmk-05-00017-t002:** Body composition characteristics.

	Visit 1	Visit 2	Visit 3	Visit 4	Visit 5	Visit 6	Visit 7	Visit 8	*p*-Value
N	14	14	13	13	11	11	17	15	
Weight (kg)	67.8 ± 12.5^1^ (54.5–95.2)^2^	67.5 ± 12.7 (54.3–95.9)	68.9 ± 13.1 (54.1–96.7)	68.9 ± 13.6 (48.5–95.9)	70.8 ± 16.5 (47.4–101.3)	70.9 ± 16.4 (47.0–98.5)	67.6 ± 12.5 (49.1–97.2)	68.4 ± 12.5 (48.8–97.5)	0.129
BMI (kg/m^2^)	24.5 ± 4.1 (20.8–33.3)	24.4 ± 4.2 (20.8–33.9)	24.7 ± 4.1 (20.6–33.5)	24.7 ± 4.5 (19.5–34.9)	25.7 ± 4.8 (20.1–35.0)	25.7 ± 4.8 (20.0–35.2)	24.9 ± 3.9 (18.7–33.6)	25.1 ± 3.9 (19.0–35.8)	0.702
LST (%)	62.3 ± 5.4 (54.5–71.9)	62.8 ± 4.4 (55.0–69.7)	62.1 ± 4.6 (54.8–71.3)	62.9 ± 5.1 (53.7–71.0)	64.0 ± 6.7 (51.9–73.2)	64.1 ± 7.2 (52.7–74.4)	61.8 ± 5.8 (53.5–71.0)	62.8 ± 5.7 (54.2–72.7)	0.291
FM (%)	34.0 ± 5.6 (24.7–42.3)	33.2 ± 4.7 (26.0–41.4)	34.1 ± 5.0 (24.3–42.2)	33.1 ± 5.4 (24.5–43.0)	33.6 ± 7.1 (24.6-44.8)	32.9 ± 7.2 (22.0–44.8)	34.4 ± 6.2 (24.6–43.6)	33.5 ± 6.5 (23.2–42.6)	0.906
VAT (g)	328.6 ± 192.0 (133.0–753.0)	310.4 ± 179.2 (129.0–688.0)	351.4 ± 207.4 (142.0–749.0)	334.7 ± 189.4 (125.0–723.0)	391.1 ± 297.5 (83.6–865.0)	358.5 ± 263.0 (77.9–793.0)	324.1 ± 187.0 (127.0–720.0)	314.5 ± 176.1 (150.0–671.0)	0.583
ASMI (kg/m^2^)	6.6 ± 0.8 (5.6–8.6)	6.7 ± 0.9 (5.6–8.8)	6.7 ± 0.9 (5.4–8.8)	6.7 ± 0.9 (5.1–8.9)	6.7 ± 0.8 (5.3–8.3)	7.0 ± 0.7 (6.4–8.4)	6.7 ± 0.6 (5.8–8.1)	6.8 ± 0.7 (6.0–9.0)	0.215
BMD (g/cm^3^)	1.1 ± 0.1 (0.9–1.3)	1.1 ± 0.1 (1.0–1.3)	1.2 ± 0.1 (1.0–1.3)	1.1 ± 0.1 (1.0–1.3)	1.1 ± 0.1 (0.9–1.3)	1.2 ± 0.2 (1.0–1.9)	1.1 ± 0.1 (1.0–1.4)	1.1 ± 0.1 (1.0–1.3)	0.414

^1^ Means ± SD. ^2^ Range. kg, kilograms; BMI, body mass index; m, meters; LST, lean soft tissue; FM, fat mass; g, grams; VAT, visceral adipose tissue; ASMI, appendicular skeletal muscle mass index; BMD, bone mineral density; cm, centimeters; Visits 1 and 5: August 2017 and 2018, Visits 2 and 6: December 2017 and 2018, Visits 3 and 7: January 2018 and 2019, and Visits 4 and 8: May 2018 and 2019.

**Table 3 jfmk-05-00017-t003:** Macronutrient intake.

	Visit 1	Visit 2	Visit 3	Visit 4	Visit 5	Visit 6	Visit 7	Visit 8	*p*-Value
N	13	13	12	12	10	8	13	14	
Kilocalories	2478 ± 748^1^ (1585–3926)^2^	2047 ± 573 (851–2894)	1890 ± 632 (929–3001)	2337 ± 937 (1171–4841)	1682 ± 533 (505–2203)	1739 ± 380 (1146–2425)	2036 ± 655 (1151–3723)	1933 ± 784 (664–3744)	0.055
Carbohydrate (g/kg)	4.4 ± 1.5 (2.3–7.6)	3.6 ± 0.9 (2.1–4.9)	3.2 ± 0.8 (2.1–4.8)	4.4 ± 1.9 (2.4–8.9)	3.2 ± 1.3 (1.6–5.6)	3.0 ± 0.6 (2.2–4.1)	4.3 ± 2.3 (1.8–10.7)	3.6 ± 2.0 (0.9–8.0)	0.455
Fat (kcal)	820 ± 279 (434–1335)	755 ± 218 (236–1022)	688 ± 282 (237–1132)	816 ± 384 (404–1795)	534 ± 200 (36–769)	569 ± 189 (240–911)	624 ± 174 (327–875)	672 ± 270 (159–1101)	0.058
Protein (g/kg)	1.4 ± 0.6 (0.5–2.6)	1.2 ± 0.3 (0.5–1.7)	1.2 ± 0.7 (0.4–2.8)	1.2 ± 0.5 (0.5–2.1)	0.9 ± 0.3 (0.4–1.4)	1.0 ± 0.2 (0.7–1.5)	1.1 ± 0.4 (0.5–2.0)	1.2 ± 0.6 (0.4–2.2)	0.093

^1^ Means ± SD. ^2^ Range. g/kg, grams per kilogram of body weight; Visits 1 and 5: August 2017 and 2018, Visits 2 and 6: December 2017 and 2018, Visits 3 and 7: January 2018 and 2019, and Visits 4 and 8: May 2018 and 2019.

**Table 4 jfmk-05-00017-t004:** Micronutrient intake.

	Visit 1	Visit 2	Visit 3	Visit 4	Visit 5	Visit 6	Visit 7	Visit 8	*p*-Value
N	13	13	12	12	10	8	13	14	
Fiber (g)	23 ± 13^1^ (10–61)^2^	17 ± 8 (7–36)	17 ± 9 (6–38)	25 ± 16 (5–60)	17 ± 11 (6–40)	15 ± 8 (8–29)	32 ± 47 (7–185)	20 ± 15 (6–66)	0.092
Vit D (IU)	132 ± 104 (4–409)	123 ± 105 (2–356)	68 ± 53 (4–177)	71 ± 56 (9–185)	67 ± 59 (15–192)	60 ± 42 (2–141)	92 ± 152 (1–551)	124 ± 241 (1–911)	0.600
Zinc (mg)	8 ± 7 (1–23)	6 ± 4 (1–13)	8 ± 5 (1–16)	7 ± 6 (0–17)	6 ± 3 (2–12)	4 ± 3 (1–10)	9 ± 6 (2–24)	7 ± 5 (1–19)	0.516
Calcium (mg)	943 ± 340 (391–1839)	814 ± 358 (215–1461)	769 ± 360 (240–1382)	869 ± 497 (122–1562)	810 ± 304 (163–1251)	693 ± 303 (266–1264)	975 ± 625 (444–2570)	749 ± 333 (282–1209)	0.377
Iron (mg)	20 ± 10 (7–37)	14 ± 10 (2–45)	16 ± 5 (8–24)	25 ± 36 (6–135)	13 ± 5 (4–20)	12 ± 6 (6–25)	18 ± 16 (6–64)	15 ± 6 (6–25)	0.498
Magnesium (mg)	169 ± 114 (31–468)	123 ± 69 (14–240)	150 ± 140 (2–547)	154 ± 158 (13–541)	142 ± 68 (32–237)	115 ± 100 (44–336)	251 ± 178 (55–737)	202 ± 162 (26–679)	0.653
Potassium (mg)	1537 ± 844 (396–2959)	1269 ± 972 (178–4158)	1386 ± 1099 (305–4527)	1479 ± 960 (181–3042)	1362 ± 788 (354–2755)	1089 ± 737 (324–2433)	2791 ± 3691 (529–1419)	1799 ± 1828 (83–7430)	0.426
Sodium (mg)	4576 ± 1275 (2401–6593)	3483 ± 1028 (1481–4886)	3383 ± 1224 (1535–5443)	3215 ± 1599 (1765–7888)	2963 ± 1306 (186–5057)	3107 ± 902 (1952–4204)	3151 ± 1653 (894–7120)	2982 ± 1073 (1223–4867)	0.161
Omega3 (g)	1 ± 1 (0–2)	1 ± 1 (0–3)	1 ± 1 (0–1)	1± 1 (0–2)	1 ± 1 (0–1)	1 ± 1 (0–1)	1 ± 1 (0–2)	1 ± 2 (0–6)	0.447
Omega6 (g)	8 ± 8 (2–29)	6 ± 5 (1–16)	7 ± 6 (1–25)	8 ± 7 (1–23)	5 ± 3 (1–10)	4 ± 3 (0–11)	6 ± 4 (1–18)	5 ± 3 (12–12)	0.091

^1^ Means ± SD. ^2^ Range. g, grams; IU, international units; mg, milligrams; Visits 1 and 5: August 2017 and 2018, Visits 2 and 6: December 2017 and 2018, Visits 3 and 7: January 2018 and 2019, and Visits 4 and 8: May 2018 and 2019.

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
