# Peer review of "Female Collegiate Dancers Body Composition, Macronutrient and Micronutrient Intake Over Two Academic Years: A Longitudinal Analysis"

_jfmk, 2020, doi:10.3390/jfmk5010017_

Round 1

Reviewer 1 Report

Dear authors,

Thank you for your considerable work. Although the manuscript discusses a high-interest topic, some issues can be addressed to improve the quality and comprehensibility:

You should replace “across” with “over” throughout the manuscript. In the Introduction section, the flow of this section is quite good. However, you mentioned that there are some college environment stressors but you did not explain what those stressors are. I think it is really important to clarify what these stressors in collegiate life are and why they could affect the body composition and dietary intake of the collegiate dancers. In addition, you should specify the “demanding environment for collegiate dancers” to facilitate the understanding of the collegiate dancers’ situation. In the Material and Methods section (Ln 99). Why did the authors decide to show VAT area instead of VAT mass? In the Material and Methods section and Abstract section. You mentioned in Ln 17-18: “Testing included anthropometrics, dual energy x-ray absorptiometry scans (Hologic Horizon™; Marlborough, MA), 3-day food logs (Food Processor, ESHA, Salem, OR) and health history questionnaires”, but you only included weight and height as anthropometric measurements. You should have included other measurements such as waist circumference, hip circumference, and waist-hip ratio. The reviewer deeply recommends you to add some measurements if you collected them. In the Material and Methods section (Ln 108), you mentioned that the used post hoc analysis was Tukey. I suppose that the authors used this analysis because of the small size of the sample. I would like to know if the authors replicated the statistical analysis with a more robust test like Bonferroni, just in case, to compare the results. If you did not, I deeply suggest doing that analysis and reporting them in case they are similar. In the Results section, you did not present any anthropometric assessments in this section. You should include BMI and weight changes. In the Results section, you should show the flow chart of the study and clarify the reasons for the drop-outs. Additionally, you must specify why if 17 participants were enrolled and completed the study you just included 17 participants in Visit 7 of Body Composition assessments. What happened in the other visits and why are there less than 17 participants included in Visit 7 dietary intake registers? In the Results section (Ln 122), you used the acronym ASMI but you did not present the meaning before. Please, check it. In the Results section, the reviewer recommends you to add a column in Table 2 for P value of variance analysis over time. In the Results section (Ln 126), Table 3 should be after the first paragraph of the Dietary Intake section. Additionally, you should include here the P values of variance analysis over time. In the Results section (Ln 145), you should include here the P values of variance analysis over time in Table 4. In the Discussion section, you said “Energetic demands fluctuate over the course of the year due to periods of increased training and performance. This requires increases or decreases in energy intake (nutrient timing) to maintain energy balance (i.e., caloric intake =  caloric  expenditure”. Did the authors register these periods of increased training in this sample or how did you find this information?

Perhaps, this sample did not change the load or frequency of trainings, and for that reason you did not find changes in your main variables.

In the Discussion section (Ln 157), you said “female collegiate dancers consumed an adequate amount of calories during some timepoints” When were that timepoints? You should specify in the Methods section. In the Discussion section (Ln 158-159), you said “This may be indicative of fluctuations in dance training, performances, or academic demands

Do you have any register of these fluctuations in trainings and academic demands?

You should include them if you have registered in order to make stronger your argument and work.

In the Discussion section (Ln 180), you wrote “time points”. You should be consistent and write every word similarly. Please, change it. In the Discussion section (Ln 193-197), you use this paragraph to justify why you decided to assess three days of the dietary intake, two weekdays and one weekend day. I think it is more properly to include this phrase in the Methods section. Please, check it. In the Discussion section (Ln 201), you mentioned that collegiate dancers trained for a minimum of 25 hours/week. However, you did not include any of these important parameters in the Methods or the Results sections. Actually, you showed in the descriptive parameters the ballet training and dance training in years, but not the current schedule of the trainings. You must include that kind of information if you have registered it.

In case you did not register that info, you should include that in the limitation section.          

In the Limitation section (Ln 2019-2020), you did not include the dietary registers of summer, winter, and spring breaks, which is a really big limitation.

Why did you not include that assessment? When were the other assessments? You must clarify this in the Methods section.

In the Conclusion section, what is the main message that you want to provide to the science community and general population? In summary, although the work presented is interesting, it needs some corrections in order to clarify better the information and message that it is sending. If the message is: “body composition and dietary intake knowledge, specifically regarding the challenges of the academic calendar”, you have to include that information to make an easier understanding of the topics to the readers. It should include some descriptive parameters, such as the hours of current training of the dancers and the academic calendar. Moreover, the design and methods should be appropriately described, and the discussion section should be revised.

Reviewer 2 Report

The aim of the study was to assess the evolution of body composition in relation to energy intake and the academic calendar of female collegiate dancers. Although this is an interesting topic and the study could bring some relevant information to the area of sport nutrition, too little information is provided. Hence I recommend the authors to revise and resubmit their manuscript with additional data and information according to the comments below

Abstract:

The description of the stat software and equipment (e.g. DXA scanner), and the statistical analysis used and is not necessary in the abstract. We should just read a short intro, study design with main techniques used, results, discussion and conclusion.

 “Calories were within recommendations” What are these recommendations? Be more specific. Is its RDA?

In the abstract, write energy intake instead of calories, protein intake instead of proteins, etc for all macro and micro nutrients.

You stats that « Habitual improper dietary intake and nutrient timing may be contributing to suboptimal BC in female collegiate dancers.” This comes as a surprise as there is no indication of improper dietary intake in the results. Please revise the results and conclusion accordingly.

Keywords:  

“body composition” is already in the title so it should be replaced by an alternative.

Introduction

The first two sentences are confusing as you first state that dance is an art and then a sport. Please clarify.

What are the health implications if adequate timing of calories and macronutrients are not met ? You could include the recent literature on REDS here and develop on the risk of low energy availability in sport.

Methods

Can you provide more information about the training load (duration, number of sessions per week, per year, intensity…) and precise when in the year the testing session took place?

What were the variables included in the anova analysis?

Results

In table 1, precise is the timing for the values reported? We assume it is pre-intervention in 2017. Be more specific.

It is surprising that fat mass be so high (>30%) for dancers. This must be discussed in the discussion;

In table 2, indicate when the visits took place?

In table 3, add an indication of training load for the weeks during which the food diaries were collected and analyzed. Indeed, Energy intake alone, with no reference to energy expenditure or at least training load is irrelevant.

In section 3.2. You refer to recommended Energy and macronutrients intake. What is it based from? Do you have a reference for this?

In the presentation of data, please add the range of values (min-max) and means ±SD for all data, micronutrients included.

Discussion

You indicate that “Energetic demands fluctuate over the course of the year due to periods of increased training and performance.” (1st paragraph). Do you have data from this study to actually show this?

It is also indicated that “In the present study, female collegiate dancers  consumed an adequate amount of calories during some timepoints, but did not meet the caloric demands during most timepoints, based on the general recommended 2,000 kilocalorie diet”. The RDA is not enough as it just valid for the general population not dancers.

L159, “This may be indicative of fluctuations in dance training, performances, or academic demands.” Can you actually describe training, performances and the academic schedule over the study period? This is crucial to discuss your results.

L183. It is indicated “appear to achieve the desired lean aesthetic through weight maintenance”. Please provide more information on this point. What are the aesthetic criteria? Can it be supported by scientific references?

The discussion is too weak. Overall we need more references specific to the sport to discuss your findings with. The training load and academic calendar of your participants must also be discussed in relation to their dietary intake. The risk of low energy availability must be discussed as well with appropriate literature.

Finally I don’t think there was a risk of malnutrition if the body composition did not change throughout the year. Besides, a fat mass value >30% is not a sign of malnutrition.

The conclusion must be reformulated as it does not reflect the findings of the study.

Round 2

Reviewer 1 Report

The reviewer really appreciates the changes did in this work.

If you have the Tukey and Bonferroni analysis, the reviewer suggests reporting in the main text the Bonferroni results, also adding a short phrase just to clarify that you did the same analysis using Non-parametric tests and results did not change.

I think that this manuscript has enough quality and it is enough interesting to the scientific and social community.

Author Response

Reviewer 1

The reviewer really appreciates the changes did in this work.

Response: Thank you very much. The authors appreciated the thoughtful feedback on the manuscript.

If you have the Tukey and Bonferroni analysis, the reviewer suggests reporting in the main text the Bonferroni results, also adding a short phrase just to clarify that you did the same analysis using Non-parametric tests and results did not change.

Response: Due to changes requested by Reviewer 2, this comment no longer applies. We re-ran the data using linear mixed-effects models since our sample size differed per visit. We eliminated the ANOVA analysis and therefore the Tukey and Bonferroni results have been removed.

I think that this manuscript has enough quality and it is enough interesting to the scientific and social community.

Response: Thank you. The authors agree that the manuscript is much improved with the reviewer edits and comments taken into consideration.

Reviewer 2 Report

The authors have responded to all the comments and provided an improved manuscript with more scientific justification, in-depth analysis of their results and more precise conclusions. 

I still have two requests before the manuscript can be accepted for publication. 

-add the years in the legend of all tables for the different visits, in addition to the month of the visits. 

-I have spotted that the number of subjects was different between the different testing sessions, which prevents you from running an ANOVA to compare the datasets. If you still want to use an ANOVA you will have to keep only the subjects who took part in all testing sessions (from visit 1 to 8). If you want to keep all the participants, you will have to use a mixed-nonlinear model to get an idea of the whole picture. This should not change too much the content of your manuscript, except that you will have to amend all tables and the results section with updated data.

Author Response

Reviewer 2

The authors have responded to all the comments and provided an improved manuscript with more scientific justification, in-depth analysis of their results and more precise conclusions. 

Response: Thank you. The authors appreciate the thorough feedback and believe it resulted in an improved manuscript

Add the years in the legend of all tables for the different visits, in addition to the month of the visits. 

Response: Thank you for noticing this error. For the tables to be stand alone, the years have been added in the footnote along with the month of the visit

I have spotted that the number of subjects was different between the different testing sessions, which prevents you from running an ANOVA to compare the datasets. If you still want to use an ANOVA you will have to keep only the subjects who took part in all testing sessions (from visit 1 to 8). If you want to keep all the participants, you will have to use a mixed-nonlinear model to get an idea of the whole picture. This should not change too much the content of your manuscript, except that you will have to amend all tables and the results section with updated data.

Response: Thank you for this comment. In order to keep all participants in the analysis the authors decided to re-run the analysis. To analyze the effects of within-semester and across semester changes we used linear mixed-effects models in program R. We also incorporated a random effect of individual ID to account for some individuals joining or leaving the study at different timepoints which is what gave us the variation in sample size between visits 1-8. Means and standard deviations remained the same at each visit however, the new analysis changed p-values reported (highlighted in the tables and text).

Round 3

Reviewer 2 Report

All corrections have been made, the manuscript can be accepted in its present form